

# Decomposability of soil organic matter over time: The Soil Incubation Database (SIDb, version 1.0) and guidance for incubation procedures

Christina Schädel[1], Jeffrey Beem-Miller[2], Mina Aziz Rad[2], Susan E. Crow[3], Caitlin Hicks Pries[4], Jessica Ernakovich[5], Alison M. Hoyt[2,6], Alain Plante[7], Shane Stoner[2], Claire C. Treat[8], Carlos A. Sierra[2]

[1]Center of Ecosystem Science and Society, Northern Arizona University, Flagstaff, AZ 86011, USA

[2]Max Planck Institute for Biogeochemistry, Hans-Knöll-Str 10, 07745, Jena, Germany

[3]Department of Natural Resources and Environmental Management, University of Hawaii Manoa, Honolulu, HI 96822, USA

[4]Department of Biological Sciences, Dartmouth College, Hanover, NH 03755, USA

[5]Department of Natural Resources and the Environment, University of New Hampshire, Durham, NH 03824, USA

[6]Lawrence Berkeley National Laboratory, Berkeley, California, USA

[7]Department of Earth & Environmental Science, University of Pennsylvania, Philadelphia, PA 19104, USA

[8]Earth Systems Research Center, Institute for the Study of Earth, Oceans & Space, University of New Hampshire, Durham, NH, 03824, USA

*Correspondence to*: christina.schaedel@nau.edu





**Abstract**


The magnitude of carbon (C) loss to the atmosphere via microbial decomposition is a function of
the amount of C stored in soils, the quality of the organic matter, and physical, chemical and
biological factors that comprise the environment for decomposition. The decomposability of C is
commonly assessed by laboratory soil incubation studies that measure greenhouse gases
mineralized from soils under controlled conditions. Here, we introduce the Soil Incubation
Database (SIDb) version 1.0, a compilation of time series data from incubations, structured into a
new, publicly available database of C flux (carbon dioxide, $CO_2$, or methane, $CH_4$). In addition
to open access, the SIDb project also provides a platform for the development of tools for
reading and analysis of incubation data as well as documentation for future use and development.
In addition to introducing SIDb, we provide reporting guidance for database entry and the
required variables that incubation studies need at minimum to be included in SIDb. A key
application of this synthesis effort is to better characterize soil C processes in Earth system
models, which will in turn reduce our uncertainty in predicting the response of soil C
decomposition to a changing climate. We demonstrate a framework to fit curves to a number of
incubation studies from diverse ecosystems, depths, and organic matter content using a built-in
model development module that integrates SIDb with the existing SoilR package to estimate soil
C pools from time series data. The database will help bridge the gap between site-level
measurements, which are commonly used in incubation studies, and global remote-sensed data or
data products derived from models aimed at assessing global-scale rates of decomposition and C
turnover. The SIDb, version 1.0, is archived and publicly available at DOI:
10.5281/zenodo.3470459 (Sierra et al., 2019) and the database is managed under a version-
controlled system and centrally stored in GitHub (https://github.com/SoilBGC-Datashare/sidb).












## 1 Introduction

Temperature, soil moisture, soil type, plant-microbe interactions, microbial community compositions, physical protection of organic matter (e.g., sorption on minerals and aggregation) and physical disconnection of microbes/enzymes and their substrates all control microbial decomposition processes and fluxes of greenhouse gases to the atmosphere (Conant et al., 2011; Schmidt et al., 2011). The relative importance of all these factors in controlling decomposition processes is poorly quantified but is important to understand as warming temperatures shift rates of microbial processes, potentially increasing releases of soil-stored C to the atmosphere (Davidson and Janssens, 2006).

Numerous reviews, syntheses, and meta-analyses have been performed using laboratory incubation studies (e.g. Conant et al., 2011; Hamdi et al., 2013; Schädel et al., 2014, 2016; Treat et al., 2015) to answer questions about the relative decomposability or stability of soil organic matter, the temperature response of soil respiration, and the ratio of $CO_2$:$CH_4$ production in anaerobic incubations. New experiments are continuously contributing to the growing body of soil incubation literature. While individual soil incubation studies are performed to answer specific research questions that may not require measuring a large variety of variables, the more details that are provided and the more comprehensive the meta-data are, the greater the utility of an individual study beyond its original use (Hillebrand and Gurevitch, 2013). Research synthesis (e.g. meta-analysis) has become an increasingly important tool in science to overcome site-specific results, identify universal patterns across ecosystems and at global scales, and to assess what is known and what needs further research (Gurevitch et al., 2018; Gurevitch and Hedges, 1999; Hillebrand and Gurevitch, 2013; Osenberg et al., 1999). Metadata help to characterize these data sets, enable finding of data through relevant criteria, and provide the information needed for data archiving (Hillebrand and Gurevitch, 2013; Jiang et al., 2015) making incubation studies as useful as possible.

Here, we report on the development and compilation of a subset of available incubation data into a new, publicly available Soil Incubation Database (SIDb). In addition to introducing SIDb, we provide clear reporting guidance for database entry and the required variables that incubation studies need at minimum to be included in SIDb. Further, we provide guidance and associated recommendations to help inform best practices for conducting consistent, comparable

soil incubation studies while retaining the adaptability required for individual research groups
and projects.

A key application of this synthesis effort is to better characterize soil C processes in Earth

system models, which will in turn reduce our uncertainty in predicting the response of soil C
decomposition to a changing climate. Soil C decomposition is most commonly represented by a
simple first-order decay function (Jenkinson, 1990) in C cycle models assuming one or more
conceptual C pools (Davidson and Janssens, 2006; Parton et al., 1987; Trumbore, 1997) with fast
and slower rates of C turnover. The models are described by several parameters such as the
decay rate of each pool, as well as the transfer rates among pools. These parameters can be
utilized to predict the evolution of $CO_2$ one would observe in an incubation over time. Incubation
time series data could therefore be used to constrain the parameters of these models by solving
the corresponding inverse problem.

We demonstrate a framework to fit such curves to a number of incubation studies from

diverse ecosystems, depths, and organic matter content using a built-in model development
module that integrates SIDb with the existing SoilR package (Sierra et al., 2012) to estimate soil
C pools from time series data. This allows users to test different model structures against their
data, representing a benefit of contributing data to SIDb. We hope the database will help bridge
the gap between site-level measurements, which are commonly used in incubation studies, and
global remote-sensed data or data products derived from models aimed at assessing global-scale
rates of decomposition and C turnover (Carvalhais et al., 2014; Koven et al., 2017). This work
also complements other compilations of soil C related datasets such as the International Soil
Carbon Network (https://iscn.fluxdata.org/), the open source Continuous Soil Respiration
database, COSORE, (https://github.com/bpbond/cosore) and the Global Database of Soil
Respiration Data, Version 4.0 (Bond-Lamberty and Thomson, 2018) and the International Soil
Radiocarbon Database (ISRaD, soilradiocarbon.org; Lawrence et al., 2019).

**2    Laboratory incubations as a tool to assess soil C decomposability**
Laboratory soil incubation studies are a commonly used method to estimate the decomposability
of soil organic matter by measuring greenhouse gas release as C is mineralized from soils under
controlled conditions. Results from incubation studies can inform global models about C pool
sizes and rates of soil organic matter processing (mostly derived from long-term incubations) and

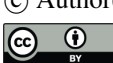

sensitivities of process rates with respect to changes in abiotic factors such as soil temperature,
moisture, pH, etc. Incubation durations may vary from less than one day to up to many years.
Short-term incubations (a few days to a few months) provide information on how much C is
readily decomposable and may be closer to the initial conditions experienced within the soil
profile. Long-term incubations (months to years) may diverge further from the conditions found
within the profile, but can give insights into the potential decomposability of slower cycling C
(e.g. Schädel et al., 2014). At the beginning of laboratory incubations, respiration of fast cycling
C dominates total C respired, but it declines rapidly, whereas slow cycling C accounts for most
of the C being respired after the fast C pool is mostly depleted (Figure 1). In this respect,
laboratory incubations serve as a method to biologically fractionate soil C into different kinetic
pools using the microbes themselves as the main fractionation agent. The time series produced is
often well approximated by a sum of exponential functions, which are the solution of systems of
first-order linear differential equations with constant coefficients (Metzler and Sierra, 2018).
Fitting data from incubations to these types of functions has been done for individual site-level
studies (e.g. Schädel et al., 2013, 2014; Sierra et al., 2017).

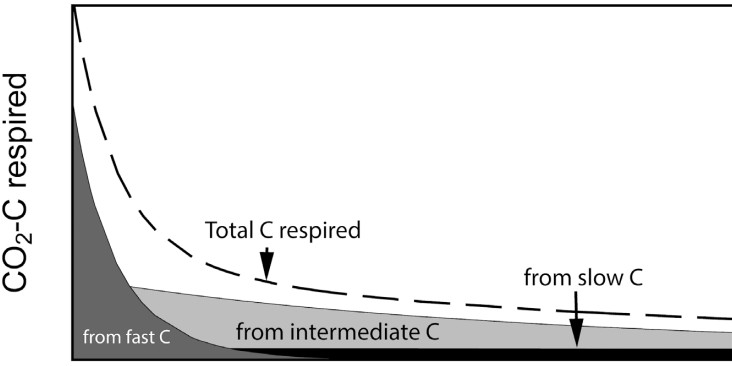

**Figure 1**: Conceptual figure of C respiration during aerobic soil incubations. Total $CO_2$-C flux is
composed of contributions from different C pools which changes over time. Fast cycling C dominates
total $CO_2$-C flux at the beginning of the incubation and is later replaced by slower cycling C pools.


Like all methods, incubations have their advantages and disadvantages. Many laboratory

methods exist for splitting soil C into pools of various purported stabilities (e.g. density



fractionation (Sollins et al., 2006), sequential extraction (Heckman et al., 2018), and thermal
analysis (Barré et al., 2016)), but incubations are the only biological assay for testing soil C
stability, an ultimately biological process. Carbon stability is a measure of how resistant and
inaccessible organic molecules are to microbial decay. Another distinct advantage of incubations
is the high level of control they allow, as compared to field methods. For example, incubations
that test the temperature sensitivity of C flux (e.g. Bracho et al., 2016; Conant et al., 2008) offer
a greater level of control compared to field measurements in several ways. First, in situ soil
respiration is a mixture of both heterotrophic microbial respiration and autotrophic root
respiration; soil incubations isolate the heterotrophic flux. Second, in situ temperatures change
daily and seasonally thereby confounding any direct effects of temperature with the phenology of
C inputs such as root exudates and litter fall. At many locations, such as those under
Mediterranean climate regimes, temperature is highly correlated with soil moisture so that the
effects of one are impossible to disentangle from the other (Sierra et al., 2015; Subke and Bahn,
2010). With incubations, temperature and moisture effects can be tested both in isolation and
with interactions. Incubations are a tractable and accessible method that can be run with minimal
equipment (scale, gas-tight jars that seal, and an $CO_2$ analyzer). Much of the utility of
incubations lies in their simplicity. Lastly, as described above, the time series data collected by
most incubations can be connected to soil C models (Sierra et al., 2012, 2014).

The main shortcoming of incubations is their isolation from the soil ecosystem.

Incubations lack new inputs, which could otherwise prime the decomposition of the existing soil
C pool (Huo et al., 2017). However, the lack of inputs simplifies the system and allows a focus
on decay processes. Substrates can be added to incubations to measure the decomposability of
specific compounds or materials (particularly if they are isotopically labeled), or to measure the
priming effect under experimentally controlled conditions, a common extension of incubation
methods (e.g. Finley et al., 2018; Pegoraro et al., 2019). Additionally, the microbial community
in incubations may not reflect in situ communities. For example, constant environmental
conditions in incubations may reduce the available niches and potentially result in a decline of
microbial diversity—an effect that has yet to be tested. The lack of inputs can also induce
changes in the microbial community as more oligotrophic microbes are favored over time.
Lastly, soils used in incubations are always disturbed to varying degrees during removal from the
field and often further in the laboratory: during sieving or root-picking procedures, or through re-



wetting prior to the start of the incubation. For example, at the time of publication, half of the
studies in our database reported sieving prior to incubation, while a third do not report pre-
incubation procedures. This disturbance may increase the susceptibility of occluded soil C to
decay via disruption of aggregates, potentially overestimating the amount of C released during
incubations relative to field conditions (Salomé et al., 2010). In general, the experimental control
of incubations allows for most of these criticisms to be explicitly tested and accounted for as
needed, and overall, the advantages of incubations far outweigh their drawbacks when the goal is
understanding C pool structure, C stability and C sensitivity to drivers such as temperature and
moisture.

## 182   3   The Soil Incubation Database (SIDb)

The Soil Incubation Database (SIDb) version 1.0 is an open source software project that provides
open access to data and is a platform for the development of tools for reading and analysis of
data as well as documentation for future use and development. The data is freely available at
DOI: 10.5281/zenodo.3470459 (Sierra et al., 2019) and the database is managed under a version-
controlled system and centrally stored in GitHub (https://github.com/SoilBGC-Datashare/sidb).

### 189   3.1 The repository

The structure of the SIDb project contains three main folders: *data*, *docs*, and *Rpkg* which
provide access to the database, the website (https://soilbgc-datashare.github.io/sidb/), and the R
package. The tree structure of the essential repository components is as follows:

```
SIDb project
Readme.md
|-- data
|-- entry1
|-- initConditions.csv
|-- metadata.yaml
|-- timeSeries.csv
|-- docs
|-- _config.yml
|-- index.html
|-- _layouts
|-- _includes
|-- assets
```





```
|-- css
|-- Rpkg
|-- DESCRIPTION
|-- NAMESPACE
|-- R
|-- man
```

**3.2 The database**

The open-source approach to SIDb allows data access, manipulation, analysis and contribution to be accomplished without proprietary software. The soil incubation data is stored in the *data* folder. Each entry in the database consists of a folder containing three files and has the name convention '*AuthornameYEAR*' (optionally with journal name abbreviation appended) and the suffix 'a' or 'b' if multiple entries for one author and year exist. 1) The *metadata.yaml* file contains the following required sections: citation and curator information, basic site information (*siteInfo*), experimental set-up of incubation (*incubationInfo*), and the metadata for the variable in the time series data (*variables*). The structure of the metadata file allows for flexible inclusion of many types of experimental and incubation designs. 2) The *initConditions.csv* file includes site, treatment, and initial soil characteristics (C content, texture conditions, etc.; Table 1). 3) The *timeSeries.csv* file contains measurements made over the course of the incubation. Column headers in the *timeSeries.csv* file are required to match the values entered for variable names in the variables section of the *metadata.yaml* file (e.g. V1:name, V2:name, etc.).

**3.2.1    The metadata file**

The metadata file is a simple text file that includes all relevant information about the incubation study. The *yaml* format is both human and machine readable. YAML (YAML Ain't Markup Language) files are text files that utilize indent hierarchy to store information in iterable and query-able format. Thus, data stored under main headings may contain subcategories and arrays of information. In an array, each line is started with a hyphen, followed by a space, then the data. A heading of any level must end with a colon, followed by a new line return. The metadata.yaml file contains four sections. The first section consists of bibliographical data about the database entry, including DOI and contact information (Fig. 2). The second section, *siteInfo*, includes geographic data, land cover, vegetation, and soil data (Fig. 3). The third section, *incubationInfo*,



provides data on laboratory experimental setup and sample treatment (Fig. 4). The fourth section,
*variables*, contains metadata for the individual columns of the timeseries.csv file (Fig. 5).

```
citationKey: # Unique identifier in the format: LastnameYearJOURNAL
doi:  # DOI of the publication where data is published
entryAuthor: # First and last name of the person who enters the data in this file
entryCreationDate: # Date when the data is entered in this file. Format: YYYY-MM-DD
contactName:  # First and last name of contact person
contactEmail: # Email of the contact person
entryNote:  # Any notes or comments related to this entry.
study:  # Overall study description
```

**Figure 2**:  Bibliographic data needed for each database entry

One advantage of the *yaml* format is the ease with which specific types of data can be grouped in
a hierarchical array. For example, in Figure 3 *site* is a subfield of *siteInfo*, and latitude is a
subfield of coordinates. More subfields can be added to the *siteInfo* subfield as necessary,
however, adding a secondary subfield beneath existing subfields should be avoided in SIDb as
consistent data structure is required for data aggregation. For example, in the *siteInfo* section, the
variables *coordinates, country, MAT, MAP, landCover*, *vegNotes* and *soilTaxonomy* all need to
be equal to the length of the site array Fig. 3.

```
siteInfo:
    site: # Names of individual sites,
        # if one site, keep on this line, if multiple, use array format
        # These fields should be arrays of equal length to site array
    coordinates:
        latitude: # Latitude in decimal units
                  #(check for negative that denotes southern hemisphere)
        longitude: # Longitude in decimal units
                   #(check for negative that denotes west)
    country: # Name of country where site is
    MAT: # Mean annual temperature in degrees Celsius
    MAP: # Mean annual precipitation in mm
    elevation: # Elevation of study site in meters above sea level
    landCover: # Land cover of the site. Valid fields are:
               # bare, cultivated, forest, rangeland/grassland,
               # shrubland, urban, wetland, tundra
    vegNote: # Additional details about land cover such as
             # species or functional type composition
    soilTaxonomy:
        soilOrder: # Soil order according to the classification system described below
        soilFamily: # Soil family description (e.g., Eutric of Eutric Cambisol)
        soilSeries: # Soil series according to the classification system described below
        classificationSystem: # Name of classification system used.
                              # Valid fields are: USDA, FAO, and WRB
    permafrost:
        permafrostExist: # Yes or blank if no (if yes, permafrost must exist at the site)
        activeLayer: # Depth of the active layer in meters
```

**Figure 3**: Site information for each database entry



In Fig. 4, the *incubationInfo* field has a subfield with a description on how the incubations were
carried out. This is important information for documenting the experimental conditions under
which the incubations were conducted.

```
incubationInfo:
    incDesc: # Short description of the incubation setup and main treatments
             # These fields should all be one dimensional arrays.
             # Values for experimental variables with multiple treatment levels
             # should be entered in the variables section, and left blank here
    depthInfo: # Soil depth in cm. If only one depth listed instead of range,
               # enter as top and bottom, 0 is organic/mineral interface.
               # If organic layer, enter 0 as top and bottom.
               # If multiple depths, leave blank and specify in variables section
      top:
      bottom:
      midDepth:
      surfaceAtm: # blank if zero is organic/mineral interface,
                  # yes if surface is atmospheric interface
      horizon: # soil horizon designation
    temperature: # Temperature at which incubations were performed in Celsius.
                 # If temperature is an experimental treatment with multiple levels,
                 # leave blank and specify in variables section
    moisture: # Use moisture as a template for any additional treatments performed,
              # i.e. report treatmentName, value, and units (if applicable)
      value: # Overall moisture at which incubations were performed.
             # If moisture is an experimental treatment with multiple levels
             # leave blank and specify in variables section
      units: # Valid fields are: percentGWC, percentFieldCapacity,
             # percentWaterFilledPoreSpace
    anaerobic: # Yes if headspace flushed with N2 or He, blank if aerobic
    gasMeasured: # Blank if CO2, other valid entries are:
                 # CH4, N2O, 13CO2, 14CO2, 13CH4, etc.
                 # Leave blank if multiple gases measured and specify in variables section
    replicates:
      value: # Number of replicates per treatment
      type: # Valid fields are: field or lab
    incubationTime: # length of incubation in days
    preincubationTime: # Pre-incubation time in days
    samplePreparation:
      intactCore: # yes or no
      sieving: # no, or mesh size in mm
      rootPicking: # yes or no
      rockPicking: # yes or no
    gasAnalyzer: # Gas analysis equipment for measurements
```

**Figure 4**: Incubation information for each database entry

The last fields that must be filled in are in the *variables* section (Fig. 5). This section consists of,
in sequential order, subsections containing the metadata that correspond to the respiration time
series observations (columns) of the *timeSeries.csv* file. The number of variables (V1-Vn) must
therefore correspond to the number of columns in the *timeSeries.csv* file. The first column in the
*timeSeries* file must be a vector of time (in days or other consistent unit), and thus the first
variable name (V1:name) in the variables section must also be "time". Experimental and
incubation treatments listed in the *incubationInfo* section must be specified under each variable
(V2, V3, etc.). Note that if a treatment has only one level it will be reported in the *incubationInfo*



section and does not need to be repeated in the *variables* section. For example, if all incubations
were conducted at the same temperature, the incubation temperature would be reported under the
*temperature* subheading in the *incubationInfo* section and the information will be automatically
propagated to all of the variables (example of Crow2019a in the database). However, if a
treatment has multiple levels, e.g. an incubation study utilizing three temperatures, the
*temperature* subheading under *incubationInfo* would be left blank, and the temperature level
would need be specified for each variable in the *variables* section in a subheading called
"temperature" (example of Bracho2018SBB in the database).

```
variables: # These describe the columns of your timeSeries.csv file
    V1: # column 1
      name: # Name of first variable in the accompanying csv data file.
            # First variable should be time
      units: # Units of first variable in accompanying file. Usually "d" for days
    V2: # column 2
      name: # Name of second variable in accompanying file
      varDesc: # Description of the variable
      site: # Site where the incubated sample was collected
      experimentalTreatment: # 'experimentalTreatment' here is a place holder.
                             # Replace this word by any of the listed variables
                             # in incubationInfo above (temperature, moisture, etc.)
                             # and type value or level after colon
      gasMeasured: # Blank if CO2, Other valid fields are:
                   # CH4, N2O, 13CO2, 14CO2, 13CH4, etc
      units: # Units in which this variable is provided if not a factor
      statistic: # Leave blank if mean values.
                 # Other valid fields include: SD, SE, and none (if a single rep)
      primaryVariableName: # Links variable with associated timeseries data
                           # collected on the same sample e.g. SD data or 13C-CO2 data
                           # associated with mean CO2 data
```

**Figure 5:** Information for each variable

### 3.2.2 Data entries

The *timeSeries.csv* file for each entry in the database contains the time series of incubation data
in comma-separated format. The first column of the data file must contain the times at which gas
measurements were taken. Subsequent columns must contain the respiration measurements. The
format of the data is irrelevant (e.g. units) as long as the relevant information to identify each
respiration column is described in the variables field of the metadata file.

### 3.2.3 The website

Documentation of the project, which includes the database and the R package, is presented on
the project's website (https://soilbgc-datashare.github.io/sidb/). The site is served at a local host



and can be viewed in any web browser. The website is publicly served by *GitHub Pages*. Every
time new changes are pushed to the SIDb repository, the website is rebuilt and served
automatically by GitHub.

**3.2.4    The R package**
Data in SIDb are stored in a format that can be read in any programming language. We provide
an R package to allow users to compile or read the database into R and a platform to facilitate
future analyses. To install the package, open R and run:

install.packages("devtools")
devtools::install_github('SoilBGC-Datashare/sidb/Rpkg/')

Two main functions are provided: *loadEntries.R* and *readEntry.R*. As their names suggest,
*loadEntries.R* collects all metadata and data from all entries and produces an 'R list' with the
entire database. The function *readEntry.R* reads individual entries from the database and also
produces an `R list`. The package also provides a function that "flattens" and coerces the
database list object into a simpler data structure for easier querying (*flatterSIDb.R*), as well as
stand-alone functions to query the entire database in its native list format for specific variables.
For instance, the function *coordinates, R* extracts all latitudes and longitudes for each entry in the
database. Similarly, other functions are provided to extract C and nitrogen (N) content, or the
incubation duration of each entry.

**3.3 Summary statistics in SIDb version 1.0**
The database is a work in progress: currently SIDb includes 31 studies with 684 time series,
representing a total number of 42,545 datapoints (Fig: 6). Most entries contain multiple time
series of $CO_2$ fluxes. Incubations reported in SIDb were performed under temperatures ranging
from 0 to 40 ˚C with the majority of incubations under normal laboratory temperature (20-25 ˚C)
(Fig. 6a). Soil temperature is the most frequently reported laboratory treatment, while soil

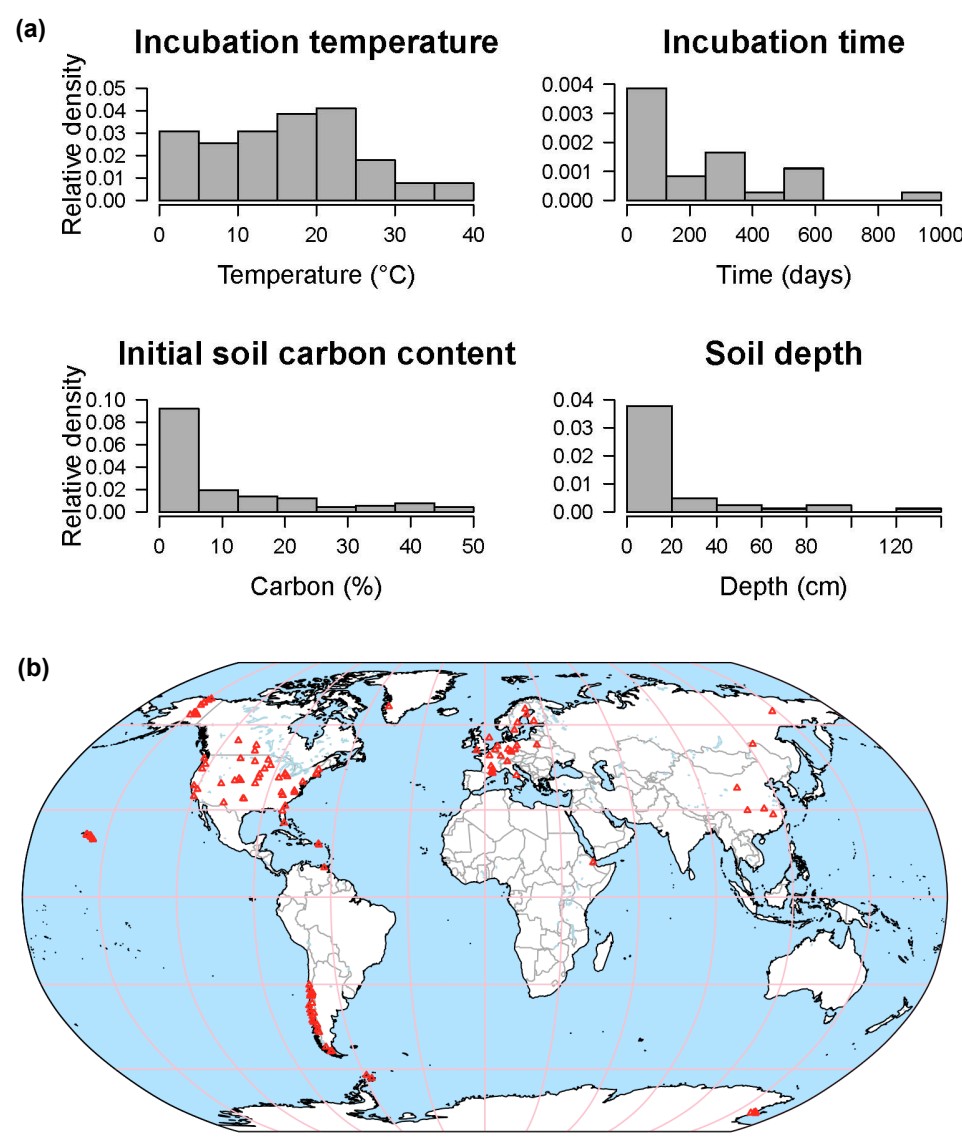

**Fig. 6** Data distribution histograms of incubation temperature, time, initial soil C content, and soil depth for available incubation data in SIDb 1.0 (a). Map of currently available incubation studies (b).

moisture is less frequently reported despite the fact that it is also a key factor in incubation
studies. The omission of soil moisture data may be related to inconsistencies in reporting
conventions, a topic that is discussed further in section 4.3. All soils listed in our database





included surface soil samples, however some studies considered soil depth as a treatment and
report incubation data from soil layers as deep as 1.2 m (Fig. 6a).

Important geographic and ecological gaps exist in SIDb version 1.0. Coverage is highest in

temperate, followed by arctic regions, with only a few studies in tropical areas while the
continents of Africa and Australia are barely represented (Fig. 6b). Incubation data from the
tropics are currently poorly represented in SIDb despite their vulnerability and the importance of
tropical regions to global C cycling, and therefore should be a priority for both future ingestion
into SIDb and further study. For most ecosystems, there are still many incubation studies to be
included into SIDb in the future. Additionally, recent work (Fontaine et al., 2007; Hicks Pries et
al., 2018; Mathieu et al., 2015) has highlighted the importance of understanding deep soil
processes and potential changes due to global warming. In fact, warming effects on respiration
have been observed at depths as great as 1m (Hicks Pries et al., 2017). Incubations of deep soils
thus represent a major gap in SIDb, which is reflective of the lack of deep soil incubation studies
more broadly, and present a large potential for future study. It was not our intention for SIDb to
introduce SIDb as a comprehensive database. Instead, we want to introduce SIDb's structure,
tools, and the current capacity of the database to the broader scientific community.

**4    Required and suggested data reporting for inclusion into SIDb**
While consistent methods across studies facilitate meta-analysis, incubation studies must remain
adaptable to each research question, available resources, and soil properties. Nonetheless, in
developing SIDb and the entry template, the most critical required components of incubations for
making comparisons across studies emerged. On the basis of these observations, we have
generated a list of variables, including information about the sites, soils, and the set-up of the
incubation itself, that we require in order for a study to be ingested in SIDb (Table 1). Here, we
discuss the issues associated with these critical variables and make suggestions for other useful
variables to report that, while not required, will increase the interpretability of results and allow
for broader inclusion into syntheses and meta-analyses (Table 1). In the supplemental material,
we also offer a limited discussion of methodologies and measurements such as incubation setup,
sample preparation, additional variables to measure, and special considerations for radiocarbon
incubations.

### 4.1 Site information

Site characteristics provide a context for the inherent conditions of the soils. General site
characteristics, such as latitude and longitude, mean annual temperature and mean annual
precipitation are important in drawing out the similarities or differences between studies.
Descriptions of the ecosystem and the aboveground vegetation give information on litter input
and chemistry, which can be a direct link to organic matter quality. Additionally, providing
information on the soil order and taxonomy helps to put findings into context with other studies
(Schimel and Chadwick, 2013).

### 4.2 Soil characteristics

There are ultimately two essential soil variables that must be reported for incubation studies, and
a myriad of suggested variables that facilitate comparisons among and explorations of potential
drivers. The first essential soil variable is depth, which is a major organizing factor of many soil
characteristics. No matter whether an individual incubation study measured soil from a single
depth increment or multiple depth increments, either the depth increment (top, bottom, and
middle) or the horizon must be reported. Ideally, both depth and horizon should be reported as
samples can be taken from a generic depth or from a mixture of horizons (when sampled to a
certain depth). All subsequent soil characteristics should then be reported for each depth
increment or horizon incubated and provided in the *initConditions.csv file*.
When reporting the sampling depth, it is necessary to report whether depth is in relation
to the soil surface, which can be defined as the top of the mineral soil or the top of the organic
horizon depending on the system, or within a specific soil horizon. Additionally, specifics of the
geography and topography of the sampling locations, such as permafrost zone, active layer
thickness, or permafrost table are crucial to report.
The second required soil variable is either the initial C (reported in mg C gdw$^{-1}$ or %) or
organic matter (which can be converted to C), which is essential for facilitating comparisons
across studies and for normalizing rates of C losses during incubations. Other common and
useful variables to measure are initial N (reported in mg C or N gdw$^{-1}$ or %), bulk density in g
cm$^{-3}$, soil texture, and pH.
Most soil characteristics, as listed in Table 1, can be measured at the beginning of an
incubation on a subsample of the soil being incubated, while others like pH, redox, or microbial



biomass may be best measured multiple times during the course of an incubation (see
Supplement for more details). For anaerobic incubations, we strongly recommend measuring
redox potential because it may not be sufficient to assume that anoxic conditions (e.g. soils
inundated with water and headspace filled with $N_2$ or He) will result in the production of $CH_4$
during the incubation as there can be a considerable lag period before $CH_4$ production occurs
(Knoblauch et al., 2018; Treat et al., 2015).

**4.3 Incubation information**
Details of incubation studies should be reported as they enhance the value of a primary study, but
also, critically, they determine whether or not they can be included in a synthesis or meta-
analysis. Thus, most of the information about how an incubation and its treatments are carried
out are required variables in SIDb. Incubation duration, temperature, and soil moisture are
among the most important details to provide because they directly affect microbial activity and
therefore C flux rates (Table 1). For temperature and soil moisture, it needs to be clarified
whether temperature and moisture were controlled at a single value or whether there were
multiple temperature or moisture treatment levels. For temperature, details on how incubation
temperature was achieved should be provided (e.g. water bath, freezer, or controlled environment
chamber). For moisture, it should be specified whether the soils were all brought to the same
moisture content or left at field conditions. For below-freezing incubation temperatures, unfrozen
soil water can also be quantified, if possible, as temperature responses of $CO_2$ production at
subzero temperatures are influenced by water availability (Öquist et al., 2009). Moisture
treatments range from fully aerobic (either drier than or at field capacity) to fully anaerobic
anoxic (headspace of jar flushed with $N_2$ or helium) to fluctuating moisture conditions. In
aerobic incubations, soils are often freely drained and deionized water is added over the course
of the incubation to maintain constant moisture content. However, caution should be paid in
order to maintain constant moisture through the incubation and not allow soils to dry out as
drying and rewetting of soils can affect C mineralization rates and microbial activity (Birch,
1958; Rey et al., 2005; Unger et al., 2010). In addition, adjustments to soil moisture are ideally
made at least 24-48h prior to making measurements to minimize confounding effects of water
addition (Rey et al., 2005). For anaerobic incubations it may not be necessary to add water
during the course of the incubation as incubation vessels typically remain closed. Other critical





parameters to report about the incubation from the synthesis perspective include whether
replicates are field (i.e., spatially different soil cores) or analytical replicates, whether soil
samples were homogenized (e.g. by soil sieving), or whether roots were removed prior to
incubation (see Supplement for more information). Lastly, the duration of a pre-incubation
should be reported if carried out.

### 4.3.1 Flux measurements

Incubation data are most commonly published as C flux rates or cumulative C release over time
for the whole incubation period. SIDb is designed around incubation studies that report
respiration rates and cumulative release over time (*timeSeries.csv*), and time series data is
required for inclusion in SIDb. Reporting only one average flux value, one maximum production
value, or one single cumulative C release value for the whole incubation period may be useful
for comparison of treatments within a study, but omits key information about changes in C
dynamics over time and precludes our ability to model dynamics of different C pools. If changes
in C dynamics over time are not of interest for a specific study, time series data should be
provided in supplementary material or in a data repository such as SIDb. Flux rates can be
provided on a per gram dry soil or per gram soil C basis, as mg $CO_2$-C g dry weight$^{-1}$ d$^{-1}$ or mg
$CO_2$-C g$^{-1}$ soil C day$^{-1}$. These units can be easily converted to one another using the required
initial C data (Table 1). Providing flux rates on a wet-weight soil basis or per volume of soil
slurry is discouraged, as SIDb does not support this format and it precludes comparisons to other
studies. If units of dry weight are not available, then soil moisture content and bulk density need
to be reported so that data can be converted to standard units. Reporting C release on a per gram
C basis captures information about C decomposability and reveals information about the relative
C release from a given soil that is independent of its C quantity; this is particularly useful for
comparisons among soils, sites and incubation studies (Schädel et al., 2014).

### 5 Case study: Fitting time series data to pool models in SIDb version 1.0

Our incubation database can be easily integrated with other R packages for further analyses. For
instance, it is possible to integrate soil C pool modeling from the SoilR package (Sierra et al.,
2012) with parameter optimization from the FME package (Soetaert and Petzoldt, 2010). We
illustrate this functionality with a simple example. The entry Crow2019a in the database contains

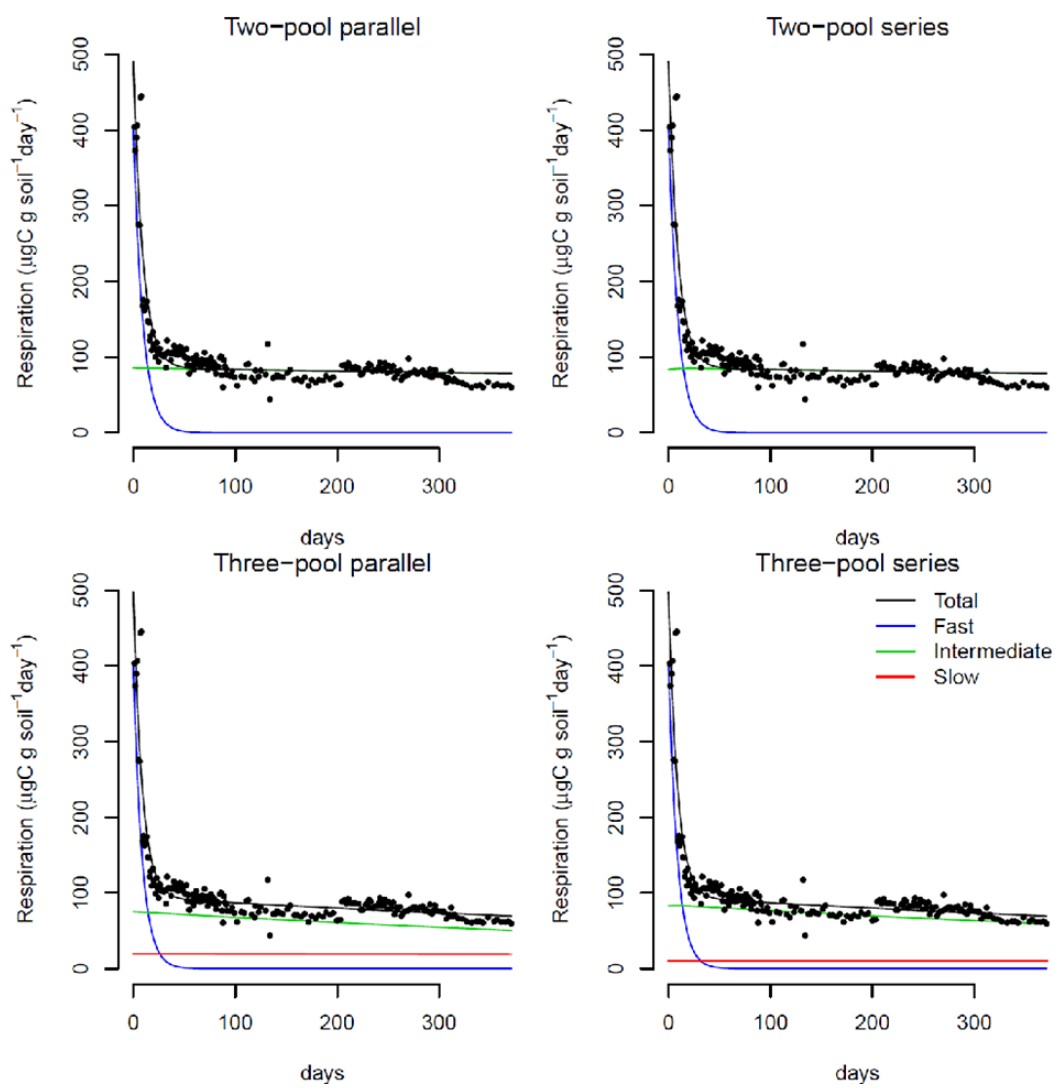

**Figure 7:** Results from a parameter optimization procedure to soil incubation data from a native tropical forest of Hawaii. The parallel model structures do not consider transfers of C among pools, while the series model structures transfer C sequentially from fast to slow cycling pools. In all cases, the models fitted the data relatively well (Table 2), and identified the relative contribution of the different pools to the overall respiration flux.

a large number of long-term incubations (371 days). From those incubations, we selected data
from a native forest in Hawaii and fitted a set of first order models with two or three pools.
Following the procedure described in Sierra et al. (2015), we optimized two- and three-pool



models with parallel, series, and feedback connections among them (Fig. 7). According to the
Akaike information criterion (AIC), the two-pool model with parallel structure is the most
parsimonious model (lowest AIC) for this specific dataset (Table 2). However, the three-pool
models show a long-term behavior consistent with our understanding of soil C dynamics (Figure
7). A parsimonious model structure that combines low AIC and theoretical understanding of soil
C dynamics would be the three-pool model with parallel structure, for which five parameters
were optimized with a reasonable mean square error and AIC (Table 2).

**6   SIDb connections to other databases**
There are two approaches to database building, which can be characterized by tradeoffs between
the scope and quantity of data, the ease of data analysis, and the simplicity of data entry. SIDb
has a narrow scope (i.e. incubation time series), allowing for the flexibility to incorporate studies
with different variable types and experimental designs, while the data itself is highly structured
in order to facilitate data analysis. Other soil databases, such as the International Radiocarbon
Database (ISRaD, Lawrence et al., 2019) or the International Soil Carbon Network (ISCN,
https://iscn.fluxdata.org/) have the advantage of a much larger quantity of data and a much
broader scope. However, maintenance and data ingestion with these larger databases becomes
much more challenging and requires either, a) relaxing control of data structure, units of
variables, and direct data oversight, such as the case with the International Soil Carbon Network,
or b) in the case of the International Radiocarbon Database, increasing the complexity of the data
structure while enforcing strict variable control, e.g. allowable names, factor levels for
categorical data, and numerical limits for quantitative data. Owing to the broader scope,
maintaining these larger databases inevitably requires additional time and effort.
However, a database is structured, establishing a common set of required measurements,
metadata, and site-level data provides transparency that helps both to identify and to reduce
systematic bias. The statistical power provided by the wealth of data points in a database such as
SIDb is only useful as long as any potential systematic bias is identified. For example, all studies
in SIDb report data at the variable level with respect to a time variable, as well as provide
information about the experimental design, where the samples were collected from, who
performed the study and how to access the original data. Additionally, providing data such as
geographic coordinates, land cover, MAT, MAP, soil taxonomy, and soil C content enables





leveraging of databases that may have a different scope but contain potentially useful supporting
data. For example, respiration time series data from SIDb could be compared to $^{14}$C content of
bulk soil or respired $^{14}CO_2$ from ISRaD (Lawrence et al., 2019) by stratifying both databases
along common variables, or a query could be made using geographic coordinates, DOI, or other
variables.

**7    Data availability**
Version 1.0 of SIDb is publicly available at DOI: 10.5281/zenodo.3470459 (Sierra et al., 2019).
Documentation of the project and the R package are presented on the project's website
(https://soilbgc-datashare.github.io/sidb/).

**8    Conclusion**
Currently, SIDb is a compilation of a wide range of incubation studies with built in capacities to
summarize the database and conduct model comparisons for fitting curves to time series data.
There is great potential benefit for the soil C community through identification and ingestion of
new datasets into SIDb. Every incubation study is planned and performed to answer a specific
question; however, when analyzed in aggregate, syntheses of incubation studies can help answer
fundamental questions about soil C pools, their stability, and vulnerability to global change.
Furthermore, setting up incubation studies involves several decision points, such as whether to
sieve or preincubate the soil, whose consequences have not yet been tested systematically, but
which may be able to be tested using SIDb.

A comprehensive collection of existing laboratory incubation data will be invaluable for

the synthesis of spatial, methodological, and functional trends, as well as for identifying key gaps
in our current knowledge. Individual researchers are encouraged to add individual study results
to the database thereby helping fill gaps in our broader understanding of soil C cycling in the
process. A key goal for the next stages of development in SIDb will focus on expanding the
geographical and ecological coverage of the entries.

SIDb is specifically designed to host incubation data with time series of respiration rates

to facilitate synthesis studies. We encourage researchers to archive their data in the format
presented here, but caution that this database is not a long-term archive. SIDb not only collects
data in a structured format, it also provides tools for data analysis and reporting through an R



package and a website. Soil incubations are a commonly used technique for answering many
different kinds of research questions, and here we provide recommendations on best practices, as
well as a common data infrastructure for reporting. We expect the size of this database to grow in
the future as it can be used as a standard repository for time series soil incubation data following
open-source standards.

**Author contribution**
C.A.S. designed the database; C.A.S., C.S., J.B.M., M.A.R., S.E.C., A.P., C.H.P., S.S., and
A.M.H. built and populated the database while J.B.M. provided technical database support. C.S.,
J.E., C.T. developed the first version of incubation recommendations and C.S. wrote up the
initial draft of the SIDb manuscript. All authors contributed to the writing.

**Competing interests.** The authors declare that they have no conflict of interest.

**Acknowledgements**
Funding for the development of the database was provided by the Max Planck Society and the
German Research Foundation (SI 1953/2-1). We also acknowledge support from the US
Geological Survey Powell Center. C.S. was supported by the National Science Foundation
Vulnerability of Permafrost Carbon Research Coordination Network grant no 955713 with
continued support from the National Science Foundation Research Synthesis and Knowledge
Transfer in a Changing Arctic: Science Support for the Study of Environmental Arctic Change
Grant no. 1331083. A.M.H., J.B.-M., and S.S. were funded by the European Research Council
(Horizon 2020 research and innovation program – grant agreement N. 695101). S.E.C. was
supported by USDA National Institute of Food and Agriculture project HAW01130-H.

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



**Table 1** Required (R) and suggested (S) variables to report and measure prior to or during time
series soil incubations.

| Variable | Unit | Time of measurement | Required/ Suggested | Notes |
|---|---|---|---|---|
| **Site information** | | | | |
| Latitude/Longitude | (decimal) degrees | A[1] | R | |
| Mean annual temperature | °C year$^{-1}$ | A | R | |
| Mean annual precipitation | mm year$^{-1}$ | A | R | |
| Ecosystem/vegetation | | A | R | descriptive |
| Soil taxonomy | | A | R | USDA, FAO, WRB |
| **Soil characteristics** | | | | |
| Horizon | | A | S | Either horizon or depth in cm is required |
| Soil Depth | | A | R | Include top, mid, and bottom of each increment incubated |
| Initial C | mg C gdw-1 or % | A | R | Initial C preferred, but organic matter allowed |
| Soil organic matter | mg C gdw-1 or % | A | R | Required if initial C not reported |
| Initial N | mg C gdw-1 or % | A | S | |
| Bulk density | g cm$^{-3}$ | A | S | |
| pH | | A, B[2] | S | |
| Soil redox potential (Eh) | mV | A, B | S | One measurement (end) or continuous. Most critical for anaerobic soils |
| Horizon texture | % clay, silt, sand | A, | S | |
| Horizon soil porosity | % (m$^3$ m$^{-3}$ x 100) | A | S | |
| Microbial biomass | mg C gdw$^{-1}$ | A, B | S | Or as mg N gwd$^{-1}$ |
| δ$^{13}$C | ‰ | A, B | S | Carbon isotope composition |
| **Incubation information** | | | | |
| Incubation duration | days | A | R | |
| Incubation temperature | °C | A, B | R | Report multiple times if not consistent |
| Incubation moisture | % | A, B | R | Gravimetric water content, field capacity |
| Temperature control method | | A | S | Descriptive; e.g. room temperature, water bath, environmental chamber |





| Variable | Unit | Time of measurement | Required/ Suggested | Notes |
|---|---|---|---|---|
| Moisture control method | | A | S | Descriptive; e.g. field conditions, added water to get to a target water content, how often checked moisture content, etc |
| Aerobic/Anaerobic | | A | R | Anaerobic if headspace flushed with $N_2$ or He |
| Treatments | | A | R | Descriptive; if quantitative include units |
| Replicates | | A | R | Field or analytical replicates |
| Sample preparation | | A | R | e.g. intact core, sieving, homogenization, roots removed |
| Pre-incubation duration | days | A | S | |
| Flux time series | mg $CO^2$-C gdw$^{-1}$ day$^{-1}$ | A, B | R | |
| Gas analysis | | A | R | Description of equipment used |

[1]A: report once
[2]B: can be reported multiple times during incubation



















**Table 2** Summary statistics from the parameter optimization procedure

| Model structure | Number of optimized parameters | Sum of squared residuals | Mean of squared residuals | AIC |
|---|---|---|---|---|
| Two-pool parallel | 3 | 113685.2 | 554.5 | -6.64 |
| Two-pool series | 4 | 113685.2 | 554.6 | -4.64 |
| Two-pool feedback | 5 | 113685.2 | 554.6 | -2.64 |
| Three-pool parallel | 5 | 109584.4 | 534.6 | -2.56 |
| Three-pool series | 7 | 109583.4 | 534.6 | 1.44 |
