# Peer review of "Decomposability of soil organic matter over time: The Soil"

_Earth System Science Data, 2019_

## Referee Comment (RC1) · Ben Bond-Lamberty (Referee) · 28 Nov 2019

General comments

===============

This manuscript describes a new Soil Incubation Database (SIDb), an effort to build a platform for current and, critically, future synthesis and meta-analysis work revolving around soil incubations. I strongly applaud this effort–we have to get away from bespoke one-off syntheses, as valuable as they are, and have these kinds of architectures going forward, so that future incubation studies can (i) be designed with these data requirements in mind and (ii) it's very easy to leverage the database. The manuscript does a nice job of describing these issues. The writing is generally clear and the ms is very appropriate for ESSD.

There are a few problems. First, the documentation of the package itself is pretty bare-bones. There's no vignette, no usage license (this is actually really important–what's on the GitHub page isn't adequate), and the built-in help files are short and not always informative.

Second, QC, of both data and code, isn't really discussed. This is something that the R package format makes both powerful and easy, but (from looking at the repository) it looks like currently you only depend on R CMD CHECK, not any custom-written tests for continuous integration. (See e.g. the SRDB repository which does this; every pull request is subjected to a battery of automated checks and the results reported on the PR page.) I strongly encourage you to think about developing some that test both the code (e.g. correct behavior of package functions) and data (e.g. QC of data entries). This would also help e.g. reviewers, of this manuscript and future efforts, to easily see correctness. For example, while I installed the package, I didn't check that its functionality was as described. You might look at the ROpenSci repositories for good examples of this.

Third, it seems like the curators of SIDb, COSORE, ISRaD, and SRDB need to talk to each other, and probably with some data specialists, to plan for interoperability and data compatibility so that future studies can make maximum (and easiest) use of these valuable data. This is a meta-issue, and not a problem with this manuscript, but worth considering. I wonder if e.g. Ameriflux/ESS-DIVE or Powell Center would support an effort like this.

In summary, this is a great effort, and a well written ms. It needs moderate revisions in a number of areas, both text and code, to maximize its clarity and utility for researchers. Thank you for your work on this!

Specific comments

================

1. Line 41: yes! Excellent

2. L. 48: site-level measurements are used in incubation studies? A bit unclear

3. L. 79-83: might move this sentence to the beginning of the paragraph

4. L. 101-103: maybe! But note Sulman et al. (2019, http://dx.doi.org/10.1007/s10533-018-0509-z) – it's not guaranteed

5. L. 111-116: great to call out these other efforts here

6. L. 144: perhaps start new paragraph for readability

7. L. 279: is served at a local host? But the URL isn't a local one. Confusing

8. Table 1 is excellent

9. Table 2 could use a few clarifying details: dataset under consideration, etc.

---

## Referee Comment (RC2) · Anonymous Referee #2 · 24 Jan 2020

The authors present a new database for soil incubation time-series experiments and an R package built for compiling and using the database. The development and compilation of the database was a considerable effort that holds promise for synthesis and meta-analysis activities that the authors hope will both improve our understanding of soil carbon decomposition dynamics and our ability to model soil carbon cycling in Earth System Land Models.

Overall, I find this effort to have been a valuable one with a useful product that warrants publication in ESSD. I find the incubation suggestions and summaries useful and appreciate the point they make regarding the importance of including additional information to increase the use of experimental results in synthesis and meta-analyses (how could anyone not provide soil moisture for an incubation experiment???). I have two major concerns: 1) that the R package does not seem to be appropriately documented or vetted (e.g., it is not in CRAN? there seems to be no package vignette or examples?) and 2) that the authors invite others to use and contribute to SIDB, which is excellent, but there is no discussion of how they will maintain (or have performed on the current version of the database) QA/QC to prevent the ingestion of incorrectly entered datasets.

Additional minor comments are below:

L156 should read "a CO2 analyzer"

L364 this statement would be stronger with examples not from permafrost (certainly there are some, at least for peatlands!)

L441 I find this example a bit too simple to be very interesting, but I am puzzled by the argument that we should chose the 3-pool model even though the 2-pool model fits the data better. I do not understand why this statement (that the 3-pool model is better) is needed at all as the point is to provide a simple example of what can be done with this type of data. It seems to me that they're approaching this example with a paradigm that 3-pool models are better than 2-pool models, and that this is a distraction from the point of the manuscript. If the authors truly think that the 3-pool model is more consistent with our understanding of soil C dynamics, they should provide more rationale, including citations from the literature. Is this not the sort of question we should be using databases like this one to revisit? Could there not be a similar argument that some kind of feedback is more realistic than a parallel structure given our current understanding of soil carbon cycling? This could either be better developed into a more interesting example and discussion regarding what we can learn from this data about soil carbon model structures or or the basics of the example should be presented without suggesting the statistically "best" model is not the one the authors

like best.

L461 delete the comma after "However" it is not needed with this usage

---

## Author Comment (AC1) · 6 Mar 2020

Response to reviewers' comments:

We thank both referees for the positive feedback and insightful comments.

Reviewer 1:

This manuscript describes a new Soil Incubation Database (SIDb), an effort to build a platform for current and, critically, future synthesis and meta-analysis work revolving around soil incubations. I strongly applaud this effort–we have to get away from bespoke one-off syntheses, as valuable as they are, and have these kinds of architectures

going forward, so that future incubation studies can (i) be designed with these data requirements in mind and (ii) it's very easy to leverage the database. The manuscript does a nice job of describing these issues. The writing is generally clear and the ms is very appropriate for ESSD. There are a few problems. First, the documentation of the package itself is pretty barebones. There's no vignette, no usage license (this is actually really important–what's on the GitHub page isn't adequate), and the built-in help files are short and not always informative.

Response:

Thank you for the positive comments and pointing out the issues with the package documentation. We previously had one vignette ("sidbQueryReportPlot"), but perhaps it was not easily visible as some methods of installing packages from github (e.g. devtools::install_github) default to "build_vignettes = FALSE".

In order to improve the documentation, we have taken the following steps: 1. Added an additional vignette, "modelFit", to demonstrate the model fitting functions built into the package. 2. Improved the documentation of most of the functions in the R package 3. Added data and entry files so that all examples are now executable. This should help clarify what the package does and what the data look like. Additionally, the files for the two entries included as examples (see sidb/Rpkg/inst/extdata/sidb_entries) also demonstrate the required directory structure for building the database in R. 4. We added a description in the main manuscript how to load the vignettes and what they do. 5. We added the user license Creative Commons Attribute 4.0 International Public License (CC BY 4.0) to the GitHub repository and added text about user guidelines and usage license in the main manuscript.

Second, QC, of both data and code, isn't really discussed. This is something that the R package format makes both powerful and easy, but (from looking at the repository) it looks like currently you only depend on R CMD CHECK, not any custom-written tests for continuous integration. (See e.g. the SRDB repository which does this; every pull

request is subjected to a battery of automated checks and the results reported on the PR page.) I strongly encourage you to think about developing some that test both the code (e.g. correct behavior of package functions) and data (e.g. QC of data entries). This would also help e.g. reviewers, of this manuscript and future efforts, to easily see correctness. For example, while I installed the package, I didn't check that its functionality was as described. You might look at the ROpenSci repositories for good examples of this.

Response:

We agree that quality control protocols are very important, and appreciate that you have brought this up. We have been using the "testthat" package and approach to code testing (see https://testthat.r-lib.org/) since the start of the SIDb project and have also implemented Travis CI for continuous integration of our code.

In response to reviewer comments we have improved and clarified our approach to QC. We summarize the changes below, but details can also be found in the document "Readme.md", located in the directory sidb/tests within the sidb GitHub repository.

Previously our test framework was located within the Rpkg directory of our GitHub repository, but we have made a few changes to our testing approach to make it more transparent. Within SIDb we consider quality control on two levels: 1. Code testing 2. Data validation

Code testing can be done both locally and remotely. For local testing we have written a shell script that runs R CMD check on the package directory (GitHub: sidb/tests/pkg_test.sh). For remote testing, we use Travis CI to run the R CMD check on the Rpkg directory of the sidb GitHub repository. This ensures that any modifications to the functions or other aspects of the sidb R package are tested every time a new commit is made in the repository, and that we will be notified of any errors, warnings, or issues.

[Figure]

We have a separate test framework for data validation. Raw SIDb data (entry files) live outside the R package in the 'data' directory. These files can be tested for conformity to SIDb standards using the file 'data_test.R' (github: sidb/tests/data_test.R). This is an R script that goes to the subdirectory 'testthat' and runs all tests that are there. Tests can be run from the command line or directly inside R using devtools. Contributors of new data must run these tests before contributing to SIDb. We will not accept pull requests with new data if at least one test fails.

We added text describing this approach briefly to the manuscript.

Third, it seems like the curators of SIDb, COSORE, ISRaD, and SRDB need to talk to each other, and probably with some data specialists, to plan for interoperability and data compatibility so that future studies can make maximum (and easiest) use of these valuable data. This is a meta-issue, and not a problem with this manuscript, but worth considering. I wonder if e.g. Ameriflux/ESS-DIVE or Powell Center would support an effort like this.

Response: We agree.

In summary, this is a great effort, and a well written ms. It needs moderate revisions in a number of areas, both text and code, to maximize its clarity and utility for researchers. Thank you for your work on this!

Specific comments ================ 1. Line 41: yes! Excellent

Response: Thank you!

2. L. 48: site-level measurements are used in incubation studies? A bit unclear

Response: We replaced site-level with point locations.

3. L. 79-83: might move this sentence to the beginning of the paragraph

Response: We have moved this sentence to the beginning of the paragraph

4. L. 101-103: maybe! But note Sulman et al. (2019, http://dx.doi.org/10.1007/s10533-018-0509-z) – it's not guaranteed

Response: We rephrased this sentence to 'Soil C decomposition is traditionally represented by a simple first-order decay function... '

5. L. 111-116: great to call out these other efforts here

6. L. 144: perhaps start new paragraph for readability

Response: we started a new paragraph for this sentence

7. L. 279: is served at a local host? But the URL isn't a local one. Confusing

Response: we will clarify this sentence

8. Table 1 is excellent

Response: Thank you!

9. Table 2 could use a few clarifying details: dataset under consideration, etc.

Response: We added 'using the database entry Crow2019a, a 371 day long incubation with soil from native forest in Hawaii' to the table caption.

Reviewer 2:

The authors present a new database for soil incubation time-series experiments and an R package built for compiling and using the database. The development and compilation of the database was a considerable effort that holds promise for synthesis and meta-analysis activities that the authors hope will both improve our understanding of soil carbon decomposition dynamics and our ability to model soil carbon cycling in Earth System Land Models.

Overall, I find this effort to have been a valuable one with a useful product that warrants publication in ESSD. I find the incubation suggestions and summaries useful and appreciate the point they make regarding the importance of including additional information to increase the use of experimental results in synthesis and meta-analyses (how could anyone not provide soil moisture for an incubation experiment???). I have two major concerns: 1) that the R package does not seem to be appropriately documented or vetted (e.g., it is not in CRAN? there seems to be no package vignette or examples?) and 2) that the authors invite others to use and contribute to SIDB, which is excellent, but there is no discussion of how they will maintain (or have performed on the current version of the database) QA/QC to prevent the ingestion of incorrectly entered datasets.

Response:

We thank the reviewer for their thoughtful comments and questions. We agree that appropriate quality control/quality assurance is necessary for both the data in SIDb and the R package.

1) While the sidb R package is not currently on CRAN, we plan to submit it to CRAN very soon. However, the version of the sidb R package that is on GitHub (<https://github.com/SoilBGC-Datashare/sidb>) has undergone thorough testing to pass all the requisite checks required by CRAN vis R CMD check.

We had one vignette bundled with the package at the time of submission ("sidb-QueryReportPlot", which demonstrates a simple work flow for compiling SIDb, reshaping the data, generating simple queries, and plotting data). We have now added a second vignette, "modelFit", to demonstrate the model fitting functions built into the package. The vignettes can be viewed on GitHub at: sidb/Rpkg/vignettes, or in R by setting the "build_vignettes" option to "TRUE" when installing the package from GitHub, e.g. devtools::install_github('SoilBGC-Datashare/sidb/Rpkg', build_vignettes = TRUE). We added text describing this approach to the manuscript.

2) In response to reviewer #1 comments, we have improved the transparency of the QA/QC process, both for code testing and for data validation. The changes are discussed in detail above, but we repeat that we have also added a document to the

GitHub repository that provides a detailed discussion of the QA/QC approach used by SIDb (GitHub: sidb/tests/Readme.md). In regards to the specific comment about QA/QC of new data, we have moved our existing test framework (which ensures new entries are correctly structured as well as checking controlled vocabulary fields) to the 'tests' directory on GitHub (sidb/tests/) so that it is easier for new users to find. Contributors of new data must run these tests before contributing to SIDb; instructions for running tests are given in the "sidb/tests/Readme.md" file mentioned previously. Our policy is not to accept pull request with new data if at least one of these tests fail. We added text describing this approach briefly to the manuscript.

Additional minor comments are below:

L156 should read "a CO2 analyzer"

Response: changed

L364 this statement would be stronger with examples not from permafrost (certainly there are some, at least for peatlands!)

Response: We added water table depth in peatlands

L441 I find this example a bit too simple to be very interesting, but I am puzzled by the argument that we should chose the 3-pool model even though the 2-pool model fits the data better. I do not understand why this statement (that the 3-pool model is better) is needed at all as the point is to provide a simple example of what can be done with this type of data. It seems to me that they're approaching this example with a paradigm that 3-pool models are better than 2-pool models, and that this is a distraction from the point of the manuscript. If the authors truly think that the 3-pool model is more consistent with our understanding of soil C dynamics, they should provide more rationale, including citations from the literature. Is this not the sort of question we should be using databases like this one to revisit? Could there not be a similar argument that some kind of feedback is more realistic than a parallel structure

given our current understanding of soil carbon cycling? This could either be better developed into a more interesting example and discussion regarding what we can learn from this data about soil carbon model structures or or the basics of the example should be presented without suggesting the statistically "best" model is not the one the authors like best.

Response:

The goal of this manuscript is to introduce SIDb, provide reporting guidance for database entry, and to show an example of how the data can be used. We agree with the reviewer that the goal of this manuscript is not to show which model is the best for a specific purpose but to provide an example how the database could be used. We removed text that was describing the best model and added text that describes that depending on the question asked criteria for the best model will be different.

L461 delete the comma after "However" it is not needed with this usage

Response: removed
* * *